# Young Pacific Male Rugby Players’ Perceptions and Experiences of Mental Wellbeing

**DOI:** 10.3390/sports7040083

**Published:** 2019-04-05

**Authors:** Caleb Marsters, Jemaima Tiatia-Seath

**Affiliations:** Te Wānanga o Waipapa, School of Māori and Pacific Studies, Faculty of Arts, The University of Auckland, Auckland 1142, New Zealand; j.tiatia-seath@auckland.ac.nz

**Keywords:** mental wellbeing, emotions, Pacific, youth, New Zealand, rugby union, rugby league, mental health, elite athletes

## Abstract

Recent studies and increased media reporting across Australasia have linked young Pacific male elite athletes to depression, suicide, and other adverse mental health-related events. Despite these accounts, little is known about the way this group experience emotions and mental wellbeing. The aim of this study was to explore young Pacific male athletes’ perceptions and experiences of emotions and mental wellbeing. This qualitative study involved 20 face-to-face interviews with young Pacific males (16–24 years) engaged in elite rugby union and rugby league programmes in Auckland, New Zealand. The results identified that athletes defined mental wellbeing in a holistic and relational manner and perceived mental wellbeing as the culmination of several interconnected factors, including: Family support, reciprocating family support, living a ‘well-balanced’ life, athletic performance, and personal development away from sports. The maintenance of a well-balanced athletic identity and positive social relations were deemed central to sustaining mental wellbeing for these young men.

## 1. Introduction

Young Pacific male athletes in New Zealand and Australia have been increasingly linked to depression, suicide, and other such adverse events as demonstrated in Australasian media, social networking sites, and recent academic studies [1,2,3,4,5,6]. While these events and the prominence of Pacific athletes in elite rugby union and rugby league suggest that mental health support should be a priority for this group, gaps remain in the literature relating to emotions and mental wellbeing among young Pacific male athletes, particularly in New Zealand.

Pacific peoples constitute 7.4 percent of New Zealand’s population [7], yet Pacific males account for just under 50 percent of all provincial rugby union players in New Zealand [8] and 42 percent of all rugby league players in the National Rugby League (NRL), an elite professional rugby league competition based in Australia and New Zealand [9]. These percentages are higher in areas with a dense Pacific population, such as Auckland, and are expected to rise as a large proportion of junior representative rugby players are of Pacific heritage [8].

Whilst this success is an achievement to be cherished and celebrated, there are many transitions and complexities associated with participation in elite youth sports and a career in professional sports. Young Pacific elite athletes are often living away from home for the first time, learning how to deal with freedom and independence, forming new social and professional relationships, making financial decisions and engaging in contractual commitments, alongside familial obligations and on-field pressures to train and perform well [2,4,10,11]. These factors can impact upon the emotions and mental wellbeing of young Pacific athletes.

Several studies have identified that elite athletes may be at increased risk of experiencing mental disorders such as depression, anxiety, and substance abuse [12,13,14,15,16,17,18]. Other studies have found that the prevalence of mental illness among elite athletes is comparable to that among the general population [19,20,21,22]. Nevertheless, the true rate of mental illness among elite athletes remains unclear given the paucity of quality research investigating the emotions and mental health of elite athletes. Furthermore, little is known about the way young Pacific male athletes perceive mental health and wellbeing. This study aimed to fill this research gap by exploring how young Pacific male athletes perceive and experience emotions and mental wellbeing as elite athletes.

## 2. Sociocultural Context

### 2.1. What is Mental Wellbeing?

The World Health Organization [23] defines mental wellbeing as a state in which an individual can realise their own potential, engage in positive relationships, be resilient in the light of typical life stresses, earn a living, and contribute to their community. It is important to note that mental wellbeing is distinct from positive affect, which can come and go, whereas wellbeing is a stable state of wellness, satisfaction, and contentment [24]. Mental wellbeing is often referred to as central to one’s overall health, but the restoration of mental wellbeing can also be a goal in itself for those experiencing mental health problems [25].

### 2.2. What are Pleasant Emotions?

Pleasant emotions are key indicators of subjective wellbeing, with higher frequencies of pleasant emotions linked to increased subjective wellbeing [26]. Understandings of pleasant emotions vary between researchers, making it a challenging term to define [18,27]. Moreover, defining pleasant emotions as universal experiences can be problematic considering the cultural assumptions and interpretations that underpin the recognition of emotions [28,29,30]. However, in general, pleasant emotions refer to pleasurable and desirable responses to a specific situation or stimuli [31]. A range of pleasant emotions have been observed in sports; for example, pride, joy, gratitude, satisfaction, hope, altruism, and love [18,27,32,33]. Historically, the literature on emotions in sports has focussed primarily on unpleasant emotions (e.g., anxiety) ahead of pleasant emotions (e.g., joy) [32]. This research explores the relationship between pleasant emotions, athletic performance, and mental wellbeing; not merely the emotion–performance relationship most commonly found in the literature [32]. This approach is based on the underlying theoretical assumption that pleasant emotions are not just momentary experiences but instead, are central to psychological growth and mental wellbeing over time [27]. Despite this conjecture, it is important to note, that unpleasant emotions can also contribute to wellbeing over time [34]. For example, initial experiences of anxiety are unpleasant but may provide valuable information about the ‘self’ and the context surrounding the self, which may facilitate psychological growth and protect mental wellbeing when similar stimuli are encountered in the future [34]. Thus, context, perception, and outcomes all influence whether emotions may be considered positive or negative with regard to mental wellbeing [34].

### 2.3. Pacific Peoples in New Zealand

‘Pacific peoples’ is an umbrella term used to group distinct ethnic affiliations and peoples that identify as belonging to one or more of the Pacific subregions of Polynesia, Melanesia or Micronesia [35]. In New Zealand, Pacific peoples account for 7.4 percent of the total population and are projected to comprise 10 percent of New Zealand’s total population by 2026 [7]. Over half of Pacific peoples, 54.9 percent, are younger than 25 years old, and most, 62.3 percent, are New Zealand-born [7,36]. The Pacific population is becoming increasingly diverse, with a large number of Pacific young people identifying with multiple ethnicities [7,36]. People with mixed, Pacific and non-Pacific, heritage also fall under the Pacific peoples umbrella [37]. Whilst Pacific peoples are commonly viewed as a homogeneous group based on some similarities, shared experiences, cultural attributes, and belief systems, each Pacific ethnic group has their own separate and unique cultural identities, languages, customs, social structures, belief systems, ideologies, histories, and worldviews [38,39]. For the purpose of this article, ‘Pacific peoples’ refers to people who self-identify as belonging to one or more of the seven largest Pacific population groups in New Zealand: Cook Islands Māori, Fijian, Niuean, Samoan, Tokelauan, Tongan, and Tuvaluan [37].

### 2.4. Pacific Male Athletes’ Participation in Elite Rugby Union and Rugby League

Rugby is the most popular sport in New Zealand, much akin to the place of American football in the United States and soccer football in European countries such as England and Spain. From grassroots to professional level, rugby is deeply engrained in both New Zealand and Pacific societies [1]. Pacific rugby players are central to the cultural and economic growth of both professional rugby codes in New Zealand and Australia, and increasingly Europe [11]. Pacific athletes have become key “commodities” in the global rugby union and rugby league labour markets, and the number of Pacific males playing professionally is expected to increase exponentially worldwide [11]. The NRL boasts a growing prominence of Pacific athletes in professional rugby league. In 1996, Pacific athletes made up only 12 percent of the NRL’s playing rosters; today, Pacific athletes make up 42 percent of the players in the NRL and over 50 percent of players in the NRL’s under-20 league—an overrepresentation compared to the relatively small Pacific populations in New Zealand and Australia [4,8,40]. Similar trends are prevalent in rugby union [11,41,42].

Pacific athletes are renowned for their natural athletic ability, unwavering determination, and robust physical traits, which contributes to why Pacific males are highly sought after in both codes of elite rugby—an argument that warrants justified critique, as it often fuels the racist stereotype that Pacific athletes are “all brawn and no brain” [41,42,43]. The growing visibility of Pacific rugby players in both codes sends a strong message that a career in professional rugby is an attractive prospect for young Pacific males financially and, for some, provides a means to validate their ‘masculinity’ and attain privileging social and cultural capital for both individuals and their families [1,41].

The rugby culture is deeply entrenched in most Pacific communities, which further enhances the appetite for young Pacific males to pursue a career in professional sports [1,44]. However, for every Pacific athlete that enjoys a long and prosperous career in professional rugby, there are scores of young Pacific males who do not advance professionally and, in some instances, experience significant challenges when transitioning away from sport [41,45]. Many stressors remain for athletes who do secure professional playing contracts, coupled with the reality that a career in either code of professional rugby is short-lived, averaging three to four years per player [46]. Alongside highly publicised expectations and familial responsibilities, this often unpredictable and high-pressure environment can hamper mental wellbeing for some athletes [1,47].

### 2.5. Pacific Mental Wellbeing in New Zealand

#### 2.5.1. Pacific Perceptions of Mental Wellbeing

Pacific perceptions of emotions and mental wellbeing offer alternative worldviews than those that dominate the mainstream literature. Without making sweeping generalisations, Western views of mental wellbeing are often biomedically framed and individualistic [37,48,49,50,51]. Pulotu-Endemann, Annandale, and Instone [50] argued that Western perceptions of mental wellbeing tend to derive from clinical perspectives that must be objective in their explanations of mental wellbeing and illness in order to facilitate standardised medical diagnoses. In most instances, this undermines the subjective nature of emotions and mental wellbeing [18,30,50]. Rather, Pacific perceptions of emotion and mental wellbeing are more holistic, based on collectivism, and where family is the foundation for individual and community wellbeing [50].

As most Pacific peoples value collectivism, it is important to acknowledge the impact of the ‘relational self’ within Pacific worldviews of emotions and mental wellbeing [48,52,53]. Bush, Chapman, Drummond, and Fagaloa [49] provide an apt description of the concept of the relational self, stating that: 

“[The relational self] is a total being comprising spiritual, physical and mental elements which cannot be separated. It derives its sense of wholeness, sacredness and uniqueness, from its place of belonging in family and village, genealogy, language, land environment and culture” (p. 142).

Pacific perceptions of emotions and mental wellbeing can generally be organised into three categories: (i) traditional perceptions; (ii) contemporary perceptions; and (iii) a blend of both traditional and contemporary views [54]. Each perspective provides insight into the way emotions and mental wellbeing may be understood by Pacific peoples as well as the diverse therein [55]. Factors relating to mixed ethnicity, cultural efficacy, religious centrality, and Pacific connectedness contribute to this milieu of worldviews, which lends itself to a wide spectrum of perceptions around emotions and mental wellbeing [56]. Whilst most young Pacific peoples in New Zealand hold contemporary perceptions of mental wellbeing, concepts of holism and collectivism remain central to Pacific youth wellbeing [57,58,59].

#### 2.5.2. The Mental Wellbeing of Young Pacific Male Athletes in New Zealand

Currently, there is a paucity of literature examining emotions and/or mental wellbeing among young Pacific male athletes in New Zealand, despite an emerging evidence base examining the lived experiences of Pacific male rugby league and rugby union players in Australia [1,2,4,5,42,60,61]. Whilst this research provides invaluable insight into the lived experiences of Pacific rugby league and rugby union players, there remains a dearth of research focussed on the mental wellbeing of these athletes.

Nevertheless, and without oversimplification, the young Pacific male athlete population falls into multiple ‘at-risk’ demographic groups experiencing unique mental health outcomes: (i) Pacific youth and (ii) Pacific males. Pacific youth in New Zealand face increased challenges in maintaining mental wellbeing, exhibiting significant resilience in the face of adversity [62,63,64]. Both Pacific males and Pacific youth experience higher rates of mental illness and are less likely to access mental health services or engage in help-seeking behaviours in comparison to other New Zealand population groups [65,66,67,68,69,70]. For example, the onset for more than half of all diagnosed mental illnesses in New Zealand occurs before the age of 18 years; 16–34-year-olds in New Zealand experience the highest rates of mental illness when compared with other age groups, and Pacific males experience the highest rates of admission to acute mental health services in New Zealand; all of which impact young Pacific male rugby union and rugby league [39,66,68,70,71]. These statistics may not be dissimilar for young Pacific male athletes, but they do provide some shed light on the mental health and emotional experiences of young Pacific males in New Zealand.

#### 2.5.3. Suicide and Young Pacific Male Athletes

Although rare, suicide is a concern for young Pacific males in New Zealand. New Zealand young people aged 15–24 years old experienced the highest rates of suicide among OECD countries in 2016, and attempted suicide rates were three times higher for Pacific youth in comparison to non-Pacific and non-Māori youth in New Zealand [65,72]. The suicide deaths of several young Pacific rugby union and rugby league players in New Zealand, Australia, and Europe suggest that these trends may be analogous among this group of athletes [1,6,73,74].

### 2.6. Summary

In summary, there is an urgent need for the exploration of the emotional and mental wellbeing experiences of young Pacific male rugby union and rugby league players in New Zealand. On this premise, this article aims to address two key components of emotions and mental wellbeing for young Pacific male athletes that are overlooked in the current literature: (i) How young Pacific male rugby union and rugby league players define mental wellbeing; and (ii) what are young Pacific male rugby union and rugby league players’ perceptions and experiences of pleasant emotions and mental wellbeing. Implications of this research for the mental health and wellbeing sector, researchers and professionals working with young Pacific male athletes in high-performance environments will also be discussed.

## 3. Materials and Methods

### 3.1. Study Design

This paper derives from a Master of Public Health project which investigated young Pacific male elite athletes’ perceptions and experiences of mental wellbeing. A qualitative methodology was utilised to explore young Pacific male rugby union and rugby league players’ perceptions and experiences of mental wellbeing at an elite level of sport [75,76].

### 3.2. Researchers Positionality

The lead researcher brought to this research an ‘insider–outsider’ perspective [77]: An insider, because the lead is a young Pacific male undertaking research with his Pacific communities, and an ‘outsider’ because he has never played sports at an elite level. Thus, although the lead researcher may be of the same heritage of study participants, he did not have complete knowledge of the subcultures and intersectional identities that exist [59,77,78]. In order to address the lead researcher’s ‘outsider’ status, the integrity of the research therein was supported by the lead researcher’s supervisor, who had previously played both codes of rugby at the elite level and is a member of an advisory group consisting of former Pacific elite athletes, national coaches, managers, psychologists, academics, and other professionals with understandings of the professional sports environment and Pacific cultures, which were constant reference points throughout this research. Their knowledge and understandings increased confidence in the credibility and rigour of the methodology and design used in this research.

### 3.3. Development of the Semistructured Interview Guide

The interview guide consisted of 11 question zones (see Appendix A). The question zones comprised semi-structured, open-ended questions to capture participants’ perceptions and experiences of emotions and mental wellbeing. The use of a semi-structured interview guide ensured consistency between interviews, without negating the autonomy of participants to openly express their personal thoughts and experiences. Adopting this approach allowed for the collection of rich and in-depth data on participants’ views and personal narratives in the context of their own lived experiences. The interview schedule was developed in consultation with experts in the field of Pacific athlete wellbeing, Pacific youth mental wellbeing, and a review of national and international evidence. 

### 3.4. Recruitment

Purposeful sampling was used during the recruitment stage. Purposeful sampling is often employed in qualitative research to ensure the recruitment of information-rich participants relevant to the phenomena of interest [79,80].

The inclusion criteria for participation were: (i) To identify with Pacific heritage; (ii) aged between 16–24 years; (iii) an Auckland resident, and; (iv) engaged in an elite rugby league or rugby union programme.

To recruit participants, an advertisement was disseminated among the researcher’s personal and professional sporting networks. Personal networks included connections via Facebook. The advertisement was also emailed to elite rugby union and rugby league organisations with a large proportion of young Pacific male athletes. These organisations showed interest in the study and shared the advertisement on their websites, among their own networks, and via social networking sites such as Facebook. This approach proved effective, as there were many ‘shares’ which broadened the advertisements’ reach among the target population.

Purposive snowball sampling was also used to expedite the recruitment process. Purposeful snowball sampling was employed by asking recruited participants to share the research advertisement with other Pacific athletes who met the study criteria. As to be expected, participants shared the advertisement with their teammates. 

Purposeful snowball sampling helped the researcher to build trust and rapport with potential participants. For example, potential participants were more likely to respond to the advertisement if they knew someone who had already taken part in an interview. This approach aligns closely with the Health Research Council of New Zealand’s [81] Pacific Health Research Guidelines, which emphasises the importance of establishing trust and positive relationships when undertaking research with Pacific communities.

A ‘Participant Information Sheet’ outlining the details of the study was sent electronically to those who responded to the study advertisement. Participation was voluntary, and each participant was given the opportunity to ask questions before arranging a convenient time and location for a face-to-face interview. At the face-to-face interviews, the study details were clarified once again before the interview commenced.

### 3.5. Participants

Twenty young Pacific males participated in this study. The average age of the young Pacific male athletes interviewed was 19.5 years. The youngest participant was 16 years of age and the eldest, 24 years of age. The ethnic makeup of participants included those who identified as Tongan (*n* = 9), Samoan (*n* = 6), Samoan/European (*n* = 1), Samoan/Niuean (*n* = 1), Cook Islands Māori (*n* = 2), and Fijian (*n* = 1). Seven participants played rugby league and thirteen played rugby union. Four participants were not contracted to any sports club, three were on age-group development contracts, seven were on semi-professional contracts, and six were on full professional contracts. A summary of participant demographic information is provided in Table 1.

### 3.6. Data Collection

The one-on-one, face-to-face interviews were undertaken in July 2016. Participants selected the time and location for their interview. Locations included local rugby clubrooms, local fast-food establishments, libraries, and participants’ homes. Interviews were carried out with 20 participants and stopped once saturation was reached and no new concepts emerged from the interviews. Interviews lasted between 45–90 min and were audio-recorded with the participant’s permission. Audio for two interviews were inaudible, which led to handwritten notes post-interview. All interviews were conducted in English.

Interviews were reminiscent of the Pacific research methodological approach known as talanoa. Talanoa is grounded on providing a safe and culturally-appropriate environment for Pacific research participants through the use of Pacific traditions and protocols during interviews to openly share their experiences [82]. During interviews, food was provided for participants and, following interviews, pākau aro’a (gifts) were given to participants as a token of appreciation for their time, knowledge, and contribution to the study. The provision of food and gifts aligns with New Zealand Health Research Council’s Pacific Health Research Guidelines and the collective Pacific values of respect, appreciation, and reciprocity [81]. The use of these Pacific research processes was central to the development and maintenance of positive and meaningful relationships with participants.

### 3.7. Data Analysis

A grounded theoretical approach was used to analyse interview transcripts and field notes obtained in the data collection phase. Grounded theory is an inductive method involving the use of constant comparative analysis to analyse data [83,84]. Constant comparative analysis utilises three stages of analysis to create emergent categories and develop theoretical models to explain the data collected and phenomena studied [85]. The grounded theory approach was deemed appropriate given the paucity of research on this topic and the need to generate new theoretical frameworks and formulate hypotheses. A key function of constant comparative analysis is that data analysis and data collection take place simultaneously, which allows for the two processes to influence one another in order to focus on and gain a deeper understanding of developing concepts [83].

The first stage of analysis in grounded theory is referred to as ‘open coding’. In this study, open coding was performed by electronically highlighting relevant and noteworthy data within each interview transcript and grouping common concepts into main categories (e.g., family support and athletic performance) and subcategories (e.g., managing relationships and goal-setting) [83]. This process was undertaken after each interview and then again after all 20 interviews were completed to ensure validity and rigour.

The second stage of analysis, known as ‘axial coding’, involved the clear partitioning of the grouped data discovered in the open coding phase data by identifying the similarities and differences between categories and subcategories [86]. Axial coding was undertaken by further developing the main categories and subcategories, as well as exploring the relationships present between the main categories and subcategories. Interview transcripts were repeatedly read for concepts that overlapped multiple categories, so they could be filed into a single category of relevant fit. This process was undertaken after each interview and then again after all 20 interviews were completed for cross-checks, validity and rigour.

The third and final stage of analysis was the ‘selective coding’ process. Selective coding involved organising the main categories and subcategories into a uniform and coherent structure to generate hypotheses regarding athletes’ perceptions towards mental wellbeing and illustrate the relationships present between these categories. The development of key themes based on participant narratives was considered the most effective approach to articulate the main categories and subcategories. This process was undertaken again after each interview and then again after all 20 interviews were completed to ensure validity and rigour.

Data analysis was repeated until saturation was reached and no new concepts emerged from the data. This allowed for strong theoretical understandings of the phenomenon to emerge [85]. All interviews were transcribed verbatim by the researcher.

### 3.8. Ethical Approval

Ethical approval was obtained from The University of Auckland Human Participants Ethics Committee, Auckland, New Zealand.

## 4. Results

Participants’ perceptions towards emotions and mental wellbeing were holistic, emphasising the importance and interconnectedness of family, friends, spirituality, sports, and a healthy lifestyle balance (see Figure 1). Athletic performance played a central role in participants’ definitions of mental wellbeing. Lastly, participants viewed personal development, both in and away from sports, as important when defining mental wellbeing.

Narratives related to holistic wellbeing and the balancing of the relational self are presented as findings under five key themes: (i) Family support; (ii) reciprocating family support; (iii) living a ‘well-balanced’ life; (iv) athletic performance; and (v) personal development.

### 4.1. Family Support

All participants described the love, support, and reassurance of family as essential elements in the cultivation of pleasant emotions and mental wellbeing. Participants agreed that having a positive relationship with family was central to living a ‘happy’ and meaningful life as well as overcoming challenges along the way. It was acknowledged that family support helped athletes to remain grounded, hopeful, and motivated them to grow and improve both on and off the field. Participants deemed these attributes essential to achieving and maintaining wellbeing and success at the elite level. The following quotes highlight the importance of family support for young Pacific male athletes: “I feel on top of the world aye, but sometimes I get a bit too overboard. So, I remember how I came from humble beginnings. It’s not the amount of stuff I have, but the love I get from family and others that makes me feel complete. It helps me keep a positive attitude and remain humble. It’s helped me grow. Everyone supporting you makes you feel good, it makes you hungry for more, and makes you want to challenge your weaknesses and improve yourself” (Participant 15). “The biggest one for me is family support. Getting that external appreciation from your family, that positive vibe coming from them, a reassuring vibe. It just gives you that happiness inside” (Participant 2).

The importance of family support and love was heightened for athletes aged 16 to 18 years, who relied heavily on their parents for support when faced with adversity such as an injury or not being selected for representative teams. For instance: “Last year in the [competition] semifinals I got concussed in the first 10 min. That was my second concussion in three weeks and it took me out of the game. I couldn’t help my team and we lost and that sort of took me out of contention for the NZ secondary schools camp. The thing that kept me going was knowing that my family was there to support me, and they kept telling me I had another year to make things right. They just kept reminding me that I’m still young and still a kid and got to have fun doing the things I do, I’ve got to enjoy yourself” (Participant 13).

### 4.2. Reciprocating Family Support

Participants stressed the importance of reciprocating family support in their perceptions and experiences of pleasant emotions and mental wellbeing. Participants found pride and joy in reciprocating the support they had received from their families. Performing well in order to make their families proud and helping their families financially were the two most common forms of reciprocity described by participants, as expressed in the following: “It’s just being able to provide [financially] for family, and just keeping the family happy. Making them proud” (Participant 4). “I would probably say being able to help, to help family and others. But only when you’re at the top of your game, because it’s hard to help someone when you’re struggling yourself” (Participant 3).

Being born in the Pacific Islands and/or being signed to a professional club was positively associated with this altruistic view towards pleasant emotions and mental wellbeing. Having family members who understood the pressures, time commitments, and mental strain associated with playing rugby at the elite level was found to be a key protective factor for mental wellbeing among athletes who emphasised the reciprocation of family support when outlining their perceptions of pleasant emotions and mental wellbeing. The following statement is an apt description of this phenomenon: “Yea [family] know how it is [being an elite athlete]. My sister use to play hockey at the top level and she tried to balance it with her studies, but she couldn’t so she knows what I’m going through and she really helps. [Family] understand the pressure and my goals and know how hard it is to get up there, because a lot of other boys are trying to chase that same jersey… Warriors, All Blacks and all that… so the support and understanding they have helps to keep me positive. And Dad’s been in my shoes, so he knows how it is from experience. So, they know, they kind of understand everything” (Participant 15).

### 4.3. Living a ‘Well-balanced’ Life

Participants defined mental wellbeing as living a well-balanced life. A well-balanced life was described as key to nurturing pleasant emotions and involved having the following factors present in one’s life: Positive relationships with family; positive relationships with friends; a positive relationship with a girlfriend or partner; performing well on the field; successfully balancing any educational or employment responsibilities; and positive spiritual wellbeing. In short, when everything is going “pretty well”. The following excerpts describe the notion of a well-balanced life for participants: “[Positive mental wellbeing] is when you’re happy. Mental, emotional, physical, spiritual, and all that. When you’re happy in all those areas you could say you are in a positive mental wellbeing… balanced. An example is when things are good with family, your girlfriend, you’re good with God, content, studies are going well, and you’re playing well. To me that’s positive mental wellbeing” (Participant 11). “I think [positive mental wellbeing] is a combination of everything. When everything off the field is going pretty well, and you’re performing well on the field, and you got a good spiritual connection at the same time. Just having all three in a good balance helps. And if one of them was a bit down it would affect other areas because they’re all connected in some sort of way” (Participant 13).

Semi-professional athletes and participants who placed a strong emphasis on athletic performance when describing pleasant emotions and mental wellbeing were less likely to comment on living a balanced life.

### 4.4. Athletic Performance

Some participants perceived pleasant emotions and mental wellbeing as a derivative of peak physical fitness and positive athletic performance, as illustrated in the following statement: “[Positive mental wellbeing] is when I’m playing well. And like we have skinfold tests and fitness tests and if my skinfold is going down then I feel real happy and feel like I’m in the best shape to play” (Participant 5).

Not all participants who prioritised their athletic performance neglected other facets of their lives, however, as described below: “[Positive mental wellbeing] is when I’m training well and at my peak and best physical wellbeing. But also maintaining a strong connection with family, having time out with the boys, and spiritually as well, being good with God. Just when everything’s going good around you and that all goes on to the field and you perform well” (Participant 16).

Overall, almost all participants considered that positive athletic performance, pleasant emotions, and mental wellbeing were intimately linked: “[Positive mental wellbeing] is just feeling relaxed. I can tell when I’m distracted or worked up about something going into a game. So my best mindset is being relaxed and enjoying myself and the company around me. Remembering we all started rugby to have fun first. So enjoying yourself and then performance and everything will come after that” (Participant 17).

### 4.5. Personal Development

Just under half of the participants acknowledged the importance of personal development in their perceptions of pleasant emotions and mental wellbeing. Participants described personal development in relation to self-improvement and their capacity to achieve new goals. Athletes defined personal development in the context of athletic performance as well as off-field activities such as education, spirituality, and self-confidence. For instance: “I reckon [mental wellbeing] is when you’re mentally prepared to do whatever you want… when you’re prepared to take risks to better yourself. So, determined and having the freedom to do want you want… like some people have boundaries that surround them, and they block themselves from doing what they want and reaching their peaks. So that self-confidence to do what you want is positive mental wellbeing to me” (Participant 14).

Participants who were contracted to a professional club were more likely to accentuate the importance of personal development for their mental wellbeing in comparison to non-contracted or semi-professional participants. Views towards personal development did not differ depending on age.

## 5. Discussion

This study entered relatively new ground, with a focus on what emotions and mental wellbeing mean to young Pacific male rugby union and rugby league players. These findings demonstrate that views towards emotions and mental wellbeing are holistic, relational, and multifaceted for this group. Findings from this study will be discussed in accordance to the following key themes.

### 5.1. Family Support

Consistent with the literature around Pacific wellbeing, a loving and supportive family was deemed central to fostering pleasant emotions and mental wellbeing for this group. This is demonstrative of the relational and collectivist nature of emotions and mental wellbeing for these athletes. Moreover, family support intensified athletes’ interest, inspiration, and determination to succeed and helped participants to remain humble and joyful; traits that participants perceived as beneficial to sustaining mental wellbeing at the elite level. Several studies affirm that found family to be a powerful motivator for Pacific rugby union and rugby league players [1,2,4,60]. In the context of emotions and mental wellbeing, this may prompt young Pacific male rugby union and rugby league players to place additional pressure on themselves to succeed professionally, which may in turn, increase the risk of unpleasant emotions such as anxiety and sadness when athletes experience failure or perceived failure [22]. Avoiding ‘letting the family down’, for example, may exacerbate unpleasant emotions such as fear of failure, which may hinder mental wellbeing and further increase negative emotions such as anxiety during stressful or challenging times; a common finding in scholarship pertaining to the lived experiences of young Pacific rugby union and rugby league players [1,2,4,5,87]. It would be beneficial to explore potential interventions for this phenomenon in future research.

### 5.2. Reciprocating Family Support

Another central aspect of mental wellbeing for these athletes was being able to reciprocate family support, in particular by making their families proud and providing financial support. This desire to reciprocate familial support is prevalent in most studies involving Pacific rugby league and rugby union players [1,2,4,5]. Again, these findings suggest that self-imposed pressures to succeed may be heightened for young Pacific athletes, as this group are likely to assume the financial responsibilities for their family [4]. The emotions attached to these pressures and obligations; namely anxiety and fear, may have long-term negative consequences for athletes’ mental wellbeing. Thus, it is vital that family support is more so visible to young Pacific athletes when they experience setbacks in their careers, such as injuries, non-selection, missing out on a professional contract or unable to provide financial support. A large proportion of Pacific athletes come from lower socioeconomic backgrounds [88], which means missing out on a lucrative professional contract can be a challenge for many Pacific athletes and their families.

### 5.3. Holistic Wellbeing and Living a ‘Well-balanced’ Life

Living a well-balanced life was an important aspect of participants’ perceptions of mental wellbeing. The connection between a balanced athletic lifestyle, having a balanced athletic identity, and positive mental wellbeing is consistent throughout the literature on athlete wellbeing [1,46,89]. In this study, participants defined a balanced athletic lifestyle as being able to partake in social activities and hobbies with family and friends; much the contrast to common thought of athletic life–balance in the literature, which is more an emphasis on non-athletic identity roles and alternative career options away from sports [90,91]. Essentially, participants’ views towards a balanced athletic identity and lifestyle prioritised the maintenance of positive familial and social relationships over the exploration of alternative identity roles and/or career options, despite many participants being engaged in some form of formal education away from sports. Again, this demonstrates the relational nature of emotions and wellbeing for these participants and the centrality of positive social relationships in relation to young Pacific athletes’ mental wellbeing. An implication of these findings is that there may be the potential to support both mental wellbeing and the development of well-balanced athletic identities by strengthening the significant value placed upon these social relationships within athletic development programmes. When looking to support the mental wellbeing of Pacific athletes, management must not only think about what is best for these athletes as individuals, but also what is best for these athletes in the context of their social and familial relationships. This is key.

### 5.4. Athletic Performance

Performing well in training and on the field is a key constituent of mental wellbeing for these athletes. Whilst most athletes agreed that performing well fostered pleasant emotions and benefitted mental wellbeing, a few participants accentuated the need to be performing well when defining mental wellbeing, suggesting that these players’ mental wellbeing and self-esteem may be heavily invested in internal and external performance appraisal. This is a concern, as previous research has linked high degrees of external performance appraisal and performance-based ‘perfectionism’ to increased mental illness among athletes [12,22,92]. Some participants centred their perceptions around emotions and mental wellbeing around performance, which could indicate that some of these athletes are committing, or have committed, to a foreclosed athletic identity without exploring, or having the opportunity to explore, other avenues away from sports [1,12,46]. This is not ideal, as athletic identity foreclosure has been found to intensify the negative emotions and psychological impact of poor performances, external criticism, injuries, and forced retirement [13,90,93]. There are many studies that explore this phenomenon among non-Pacific athletes, and it would be useful to identify effective strategies to deter the development of foreclosed identities among young Pacific athletes in future research.

### 5.5. Personal Development

It appears evident that participants valued their autonomy and the opportunity to improve themselves on and off the field. Formal education was believed by participants as an important towards their personal development. Participants admitted that their parents and team staff constantly reinforced the need to obtain a tertiary degree or apprenticeship, which encouraged them to take their education seriously. These findings are positive and imply that current strategies to promote ‘back-up’ career options and develop balanced athletic identities may be working effectively; once again, reinforces the benefits of including Pacific players’ parents and family in overall player welfare and mental wellbeing initiatives. 

In the context of mental wellbeing, this finding is important, as personal development has been associated to reduced depressive moods and improved self-esteem and self-confidence for young males and young athletes alike [22,94,95]. These findings highlight the potential to foster pleasant emotions and support mental wellbeing through personal development initiatives that build upon Pacific rugby players’ existing interest in improving themselves on and off the field.

It is crucial to recognise that personal development is not inherently fostered through participation in sports, and therefore key, that elite sports programmes have initiatives in place to facilitate personal development for young athletes [96]. Finding ways to effectively develop the life skills required to succeed in a career at the elite level is also an area that requires further investigation.

### 5.6. Emotions, Cultural Identity and the Athletic Sense of Self

There was a strong association between pleasant emotions, cultural identity, and athletic identity for participants in this study. Many factors conducive to pleasant emotions and mental wellbeing were rooted in participants’ intersecting identities as young Pacific men and elite athletes. Traditional perceptions of what it means to be a Pacific man, for example, are measured by one’s ability to contribute financially, to support others, and to make their families proud; attributes that were recurrent in participants’ understandings of pleasant emotions and mental wellbeing [97]. Other factors central to participants’ perceptions of pleasant emotions, such as performing well and personal development, similarly align with the athletic sense of self and what it means to be a successful elite athlete. The literature on emotions affirms that there is no aspect of emotion that is independent of culture, so it is important that potential initiatives and interventions acknowledge the influence of these cultural markers on emotional response and mental wellbeing for Pacific athletes [18,29,30,31]. 

### 5.7. Hypermasculinity, Mental Wellbeing, and the Suppression of Emotional Expression

Gender socialisation contributes substantially in the way young Pacific male athletes express emotions, view mental wellbeing, and construct their masculine and athletic identities [98,99,100]. Hypermasculine attitudes towards Pacific male athletes were a contributing factor to the stigma participants attached to emotional expression, mental wellbeing, and mental illness [10]. The hypermasculine environment of elite rugby union and rugby league often encourages athletes to deny ‘weakness’ and supress emotional expression in order to display emotionless traits indicative of a ‘mentally tough’ athlete [12,98]. These hypermasculine views were common among the young Pacific male athletes interviewed, with some athletes believing that there is no room as young Pacific men to overtly display certain emotions, such as fear, sadness, and anxiety, which may be perceived as ‘weak’ within the elite sports environment and among their peers. Participants stated that these hypermasculine expectations are a major contributor as to why most young Pacific athletes tend to keep mental health issues to themselves [10]. These findings reinforce the evidence, with Horton [1], Hokowhitu [101], Uperesa [102], and Teaiwa [47] highlighting that the hypermasculine stereotypes placed upon Pacific male athletes are often internalised, embodied, and become a part of Pacific male athletes’ identity and psyche. Teaiwa [47] illustrates that Pacific male athletes are at the forefront of elite rugby and marketed in a hypermasculine manner through the use of terms like ‘warriors’ and ‘beasts’, which both glorifies and demonises athletes’ masculine and athletic traits as ‘primitive’.

Whilst these hypermasculine traits can be beneficial in the world of elite sports and are held in high esteem within some Pacific cultures, they can be harmful when it comes to emotional expression, mental wellbeing, and help-seeking for mental health challenges [18,32,33,47,103]. Recent studies assert that hypermasculine views towards emotional expression and mental wellbeing often increase internalised stigma towards depressive moods, restrict ways of coping, and promote the masking of emotions, all of which increase the risk of self-harm and suicide [1,12,104]. Placing less emphasis on the limiting hypermasculine caricatures that currently underpin what it means to be a Pacific male athlete and promoting culturally appropriate value-based traits, such as altruism, resilience, support, and service, could have widespread benefits for how young Pacific male athletes view their own masculinity and their attitudes towards emotional expression, mental wellbeing, and help-seeking [47,102].

### 5.8. Strengths, Limitations and Areas for Future Research 

This study is the first qualitative investigation in New Zealand examining young Pacific male athletes’ definitions, perceptions, and experiences of pleasant emotions and mental wellbeing.

The findings of this study cannot be generalised for all young Pacific male rugby union and rugby league players given the sample size, and the views expressed by these participants may not be representative of the wider young Pacific male athlete population in New Zealand. Increasing the sample size and extending the recruitment location would address this limitation and could possibly result in different findings.

The voluntary nature of this study may also reduce the representation of participants, as self-selection bias may be present. Again, further research with a greater sample size may mitigate this limitation.

Whilst the literature and findings from this research confirm that the Pacific identity is prominent among Pacific athletes in New Zealand, ethnic-specific research should be undertaken in the future to improve the rigour and applicability of research findings in this topic area.

It would be beneficial to undertake a similar study involving older or retired Pacific male athletes, as their narratives and experiences may differ but contribute significantly to the wellbeing of younger generations of Pacific male athletes.

Throughout the research process, there was strong feedback from Auckland’s sporting community to undertake a similar research project focussed on the wellbeing of female Pacific athletes. The findings from such a project would differ significantly to the findings of this research, given the unique sociocultural experiences of female Pacific athletes.

### 5.9. Implications

This study is the first qualitative investigation in New Zealand examining young Pacific male athletes’ definitions and perceptions of mental wellbeing.

Young Pacific athletes need access to clearly communicated information about how to stay mentally healthy, how to understand and recognise mental health problems, and where to go to get help. In other words, more efforts must be afforded to raising mental health literacy—preferably, as soon as they enter elite age group programmes, better still, in early education.

Athletes noted a significant disconnect between themselves and the mental health services available. There is a need to further develop and implement gender-specific and culturally appropriate and competent mental health services aimed specifically at engaging and meeting the mental health needs of young Pacific male athletes [10]. Collaborating with Pacific sports networks, local clubs and schools, and Pacific community groups would be a worthwhile step towards improving Pacific engagement with these services.

With any initiative or clinical interaction, it is essential to maintain an appropriate balance between clinical, social, spiritual and cultural perspectives of mental health to ensure efficacy and meaningful engagement with Pacific athletes, families, and communities.

## 6. Conclusions

Young Pacific male rugby union and rugby league players view mental wellbeing as the culmination of several interconnected factors. For these young Pacific male athletes, if one factor suffers, then their emotions and mental wellbeing as a whole suffer, so it is particularly important that athletes are supported during these times and that the ‘balance’ between these factors be restored. The holistic and relational way participants perceived emotions and mental wellbeing highlights the importance of familial servitude and social belonging for these athletes and reaffirms the existing literature around Pacific mental health [48,49,52,53,55]. Whilst participants prioritised a balanced athletic lifestyle, their experiences of pleasant emotions and mental wellbeing, away from family and social relationships, were largely centred around sports. This suggests that some of these athletes may be at increased risk of developing overly salient, or foreclosed, athletic identities [1,12,46]. From the findings of this study and the limited literature in the field, the development of balanced athletic identities that prioritise sports, education, strong cultural ties, and positive social relationships is crucial to fostering pleasant emotions and supporting mental wellbeing among young Pacific male athletes participating in elite rugby union and rugby league.

## Figures and Tables

**Figure 1 sports-07-00083-f001:**
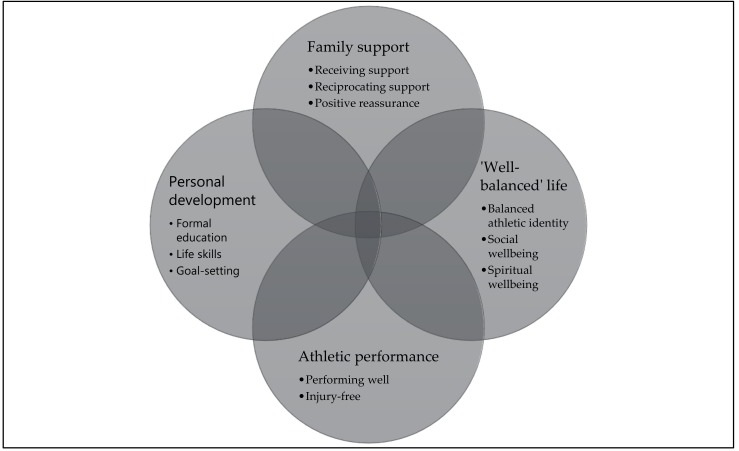
A graphical summary of the factors participants perceived as most conducive to mental wellbeing as elite athletes.

**Table 1 sports-07-00083-t001:** Demographic summary of participants.

Participant	Age	Ethnicity	Code	Contract
1	22	Tongan	Union	None
2	22	Samoan	League	Semi-professional
3	18	Samoan	Union	Semi-professional
4	19	Tongan	Union	Professional
5	19	Tongan	Union	Professional
6	23	Tongan	League	None
7	23	Fijian	League	None
8	19	Samoan/ European	Union	Professional
9	19	Tongan	Union	Professional
10	16	Cook Islands	Union	None
11	21	Samoan/Niuean	League	Professional
12	16	Tongan	League	Development
13	17	Samoan	Union	Development
14	18	Tongan	Union	Development
15	19	Samoan	Union	Semi-professional
16	18	Samoan	Union	Semi-professional
17	19	Tongan	Union	Professional
18	19	Tongan	Union	Semi-professional
19	24	Samoan	League	Semi-professional
20	19	Cook Islands	League	Semi-professional

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
