# Peer review of "Young Pacific Male Rugby Players’ Perceptions and Experiences of Mental Wellbeing"

_sports, 2019, doi:10.3390/sports7040083_

Round 1

Reviewer 1 Report

The article is very interesting and concise. It deepens in a topic of interest and understudied such as that of male athletes’ perceptions of mental wellbeing. Before accepting it for publication I would like the authors to address some minor concerns.

Please do not be scared by the amount of points I am making, as all of them are mostly minor remarks and it is my hope that they will help to improve the manuscript

1)      Given the focus of the study on rugby players it will be probably better to make it more explicit in the article’s title. Someone may expect to read about different types of athletes with the current title (although this is only a suggestion).

2)      I am not familiarize with sports in Australasia. Rugby is probably the most visible and prominent sport in New Zealand as soccer is in Europe. From this point of view, it will be good to include a rationale for the focus on rugby players in order that the international audience of the journal understand the selection of this concrete sport.

3)      In the participants sections the sentence “the average age of the participants was 19.5 years” is written twice. Please, delete one.

4)      In relation to the interviews; did the data collection ended when participants did not longer reveals new patterns, themes or other findings?

5)      Data analysis require a more detailed description to allow readers to follow each step of the analysis. For example, did the authors used triangulation of data in the categorization procedure? Please, explain how triangulation was assured in each phase described.

6)      Results are clear and easy to follow. Indeed, there are really interesting and offer a much need feedback to understand wellbeing among male rugby players. However, I found that in the results analysis there is a lack of criticality. A convincing analysis should explicitly seeks out and test rival explanations and counter examples. Were there any discrepancies between the participants that are important to signal?

7)      In the discussion I have missed a deeper analysis of how gender socialization (masculinity) could be related with the way male rugby players build their identity. It is briefly mention by authors in “the athletic sense of self” but it will be great to add a gender perspective to the discussion.

Author Response

Response to Reviewer 1 Comments

Point 1: Given the focus of the study on rugby players it will be probably better to make it more explicit in the article’s title. Someone may expect to read about different types of athletes with the current title (although this is only a suggestion).

Response 1: Article title changed to mention rugby players

Point 2: I am not familiarize with sports in Australasia. Rugby is probably the most visible and prominent sport in New Zealand as soccer is in Europe. From this point of view, it will be good to include a rationale for the focus on rugby players in order that the international audience of the journal understand the selection of this concrete sport.

Response 2: Sentence added to the start of paragraph 2.3, line 78, explaining the popularity of rugby union and rugby league in New Zealand. “Rugby is the most popular sport in New Zealand, akin to the place of gridiron football in the United States and soccer football in European countries such as England and Spain. From grassroots to professional level, rugby is deeply engrained in both New Zealand and Pacific societies.”

Point 3: In the participants sections the sentence “the average age of the participants was 19.5 years” is written twice. Please, delete one.

Response 3: Deleted repeated sentence

Point 4: In relation to the interviews; did the data collection ended when participants did not longer reveals new patterns, themes or other findings?

Response 4: Added a sentence in line 244: “Interviews were carried out with 20 participants and stopped once saturation was reached and no new concepts emerged from the interviews.”

Point 5: Data analysis require a more detailed description to allow readers to follow each step of the analysis. For example, did the authors used triangulation of data in the categorization procedure? Please, explain how triangulation was assured in each phase described.

Response 5: Explained how data analysis was undertaken and what actions were taken to ensure validity and rigour during the data analysis process. “This process was undertaken again after each interview and then again after all 20 interviews were completed to ensure validity and rigour.”

Point 6: Results are clear and easy to follow. Indeed, there are really interesting and offer a much need feedback to understand wellbeing among male rugby players. However, I found that in the results analysis there is a lack of criticality. A convincing analysis should explicitly seeks out and test rival explanations and counter examples. Were there any discrepancies between the participants that are important to signal?

Response 6: This feedback really helped to develop the findings section. I added in additional information and some new quotes to the findings (e.g., differences in the findings between age groups and playing level). New sentences added to lines: 321-329; 341-353; 373-374, 387-388, 406-408.

Point 7: In the discussion I have missed a deeper analysis of how gender socialization (masculinity) could be related with the way male rugby players build their identity. It is briefly mention by authors in “the athletic sense of self” but it will be great to add a gender perspective to the discussion.

Response 7: Again, this feedback really helped to develop the article. A new discussion section was added to paragraph 5.7, lines 498-525.

Reviewer 2 Report

This paper presents a qualitative study of perceptions of mental well-being in a sample of young adult male athletes from a Pacific Island background in New Zealand. Strengths of this study include its strong conceptual background, very well-constructed qualitative methodology, and its focus on an under-served population at significant risk for negative mental health outcomes. Overall, this is a strong contribution, although I have a few comments which may help to broaden the discussion.

Although it is beyond the scope of the present study, it would be instructive to briefly discuss what your findings might suggest for other overlapping populations -- for example, young female Pacific athletes, Pacific athletes (or former athletes) in older age groups, etc.

A brief summary of directions for future research and any potential clinical implications would be helpful.

Author Response

Thank you for your feedback. Please find the responses below.

Response to Reviewer 2 Comments

Point 1: Although it is beyond the scope of the present study, it would be instructive to briefly discuss what your findings might suggest for other overlapping populations -- for example, young female Pacific athletes, Pacific athletes (or former athletes) in older age groups, etc. brief summary of directions for future research and any potential clinical implications would be helpful.

Response 1: The literature suggests that there would be little similarity between the experiences of young male Pacific athletes and young female Pacific athletes. Mostly because of the type of sports played and the differences in sociocultural experiences. I have added a note highlighting the need for more gender-specific research on the experiences of young female Pacific athletes (see lines 543-546). I also highlighted the potential benefits of undertaking similar research with older or retired athletes, as their wisdom and experiences may be beneficial for better supporting future generations of Pacific athletes (see lines 437-439 and 540-543).

Point 2: A brief summary of directions for future research and any potential clinical implications would be helpful.

Response 2: Alongside the additions pointed out in response 1, I added more details on the potential clinical implications of this research. Particularly around how to improve mental health service engagement and clinical engagement (see lines 554-562).
